# MANIFOLD K-MEANS WITH $\ell_{2,p}$-NORM MAXIMIZATION

## ABSTRACT

Although a variety of different methods have emerged in the field of clustering, K-means still occupies an important position, and many advanced clustering methods even rely on the K-means to achieve effective cluster detection. However, the sensitivity of K-means to the selection of the initial cluster center and its limited ability to handle nonlinear separable data somewhat restrict its clustering performance. In order to overcome the limitations of K-means, we draw inspiration from manifold learning and redefine K-means as a manifold K-means clustering framework. This framework supports various types of distance matrices, thus facilitating the efficient processing of nonlinear separable data. A unique advantage of this approach is that it does not require the calculation of the cluster center, while it maintains the consistency between manifold structure and cluster labels. Additionally, we highlight the significant role of the $\ell_{2,p}$-norm; by maximizing the $\ell_{2,p}$-norm, we can ensure the balance of classes in the clustering process, which is also supported by theoretical analysis. The results from extensive experiments across multiple databases substantiate the superiority of our proposed model.

## 1 INTRODUCTION

In the field of data analysis and pattern recognition, clustering is an unsupervised learning method aimed at grouping data points into clusters. The objective is to ensure that data points within the same cluster exhibit high similarity, while those from different clusters show significant differences. This methodology has garnered considerable attention over the past decades, leading to the development of various clustering algorithms designed to enhance data annotation and parsing. Among these, K-means is particularly notable for its popularity. It partitions the data into $K$ clusters and iteratively optimizes the centroids to minimize the intra-cluster sum of squared distances. K-means is celebrated for its simplicity, intuitiveness, and efficiency, making it a staple in many applications.

Despite its prominence, K-means is not without drawbacks. Specifically, it uses metrics like the Euclidean distance to assign data points to the nearest cluster center, which may not be effective for data with nonlinear distributions. To address this, researchers employ kernel functions—such as the linear kernel Vankadara & Ghoshdastidar (2020), Euler kernel Lin & Chen (2023a), and multi-kernel Liu (2023); Yao et al. (2021)—to map data into a high-dimensional kernel space where it becomes linearly separable. Nevertheless, even with kernel-based enhancements, the choice of initial cluster centers remains crucial in K-means. Incorrect initial selections can lead the algorithm to converge to local optima, significantly impacting the clustering outcome Peña et al. (1999); Li et al. (2021); Xiong et al. (2016); Xie et al. (2020); Liang et al. (2024).

To mitigate this issue, various strategies for selecting initial clustering centers have been proposed. These include K-Means++ Bachem et al. (2016); Arthur & Vassilvitskii (2007), which reduces randomness; methods based on data point density Lan et al. (2015); spatial distribution-based partitioning techniques Aslam et al. (2020); and genetic algorithms that optimize for high-quality centers Laszlo & Mukherjee (2006). Additionally, some studies combine spectral clustering with K-means to bypass the need for estimating the centroid matrix. Since spectral clustering can automatically determine the initial cluster center, it also alleviates the sensitivity of K-means to initial center selection Pei et al. (2023). However, these approaches often overlook the alignment of data geometry with labels, which can limit their effectiveness on complex datasets.

Inspired by manifold learning Roweis & Saul (2000); Belkin & Niyogi (2001), we develop a new clustering algorithm. Manifold learning effectively captures and retains the complex nonlinear

geometry inherent in data Cai (2015); Wu et al. (2022). From this perspective, our algorithm reinterprets and enhances the traditional K-means algorithm by directly estimating the data clusters, avoiding the need to estimate a centroid matrix, and utilizing manifold learning to accurately capture the geometric structure of the data.

It is noteworthy that during the algorithm design process, we also incorporate the concept of $\ell_{2,p}$-normWang et al. (2018); Zhao et al. (2024). $\ell_{2,p}$-norm minimization often plays a crucial role in areas such as image recovery, text compression, and signal processing. However, our study also reveals additional significant roles for $\ell_{2,p}$-norm. Specifically, by maximizing the $\ell_{2,p}$-norm, we ensure the balance of classes in the clustering process, thus avoiding the problem that some classes are too large or too small, and enhancing the effectiveness and stability of clustering.

Specifically, our method differs from existing K-means techniques in several key aspects:

- We establish the link between K-means and manifold learning, ensuring the consistency of manifold structures and clustering labels.
- Unlike traditional K-means and its variants, our approach directly obtains data clusters without the need for estimating cluster centers.
- We identify a significant role of the $\ell_{2,p}$-norm. By maximizing the $\ell_{2,p}$-norm, we can ensure balanced clusters in the clustering process and provide a theoretical analysis.
- A unified clustering framework is established, and by utilizing different distance functions (such as Euclidean distance, kernel Euclidean distance, KNN distance, etc.), we can derive various K-means variants.

## 2 RELATED WORK

The essence of traditional K-means clustering is to find a set of cluster centers and then assign data points to the nearest cluster center using Euclidean distances. However, when the data is nonlinearly separable, K-means may not accurately reflect the similarities and differences between the data.

To address nonlinear separability in the original feature space, an efficient method is to employ the kernel trick. This technique maps the raw data into a high-dimensional feature space where the features become linearly separable. This approach has inspired the development of kernel K-means and various variants. Girolami Girolami (2002) pioneered the integration of clustering and kernel methods by proposing a clustering method based on the Mercer kernel. Kong et al. Kong & Kong (2013) used a conditional positive definite kernel (CPD) to map data into a high-dimensional space and performed K-means clustering there. Wu et al. Lin & Chen (2023a) introduced the Euler kernel to kernel K-means by mapping input data onto a unit hypersphere in an equal-dimensional space and performing K-means clustering on that sphere. However, Lin et al. Lin & Chen (2023b) noted that the center of mass tends to deviate from the surface of the unit hypersphere during Euler kernel clustering, leading to outliers. To address this issue, they proposed constraining the center of mass to the unit hypersphere or optimizing the mapping of the original data in Euler kernel space, effectively handling the problem of center of mass deviation.

However, the performance of kernel K-means clustering largely depends on the choice of kernel functions. To alleviate this issue, multiple kernel learning has been introduced into K-means clustering to find the best kernel combination for clustering. Liu et al. Liu et al. (2017) proposed an adaptive optimal neighborhood multi-core clustering model, which employs matrix-induced regularization to enhance the diversity of selected kernels and the representability of optimal kernels. In contrast, Yao et al. Yao et al. (2021) enhanced kernel diversity from the perspective of subset selection by choosing representative kernels from predefined sets. However, both approaches Liu et al. (2017); Yao et al. (2021) rely on additional discretization steps to obtain the final discrete clustering indicator matrix. To bypass this discretization step, Wang et al. Wang et al. (2022) proposed a discrete and parameterless multi-core k-means model. By implicitly introducing regularization terms to assess the correlation between different kernels and using alternative optimization methods, this model directly generates the cluster index matrix without further processing.

It should be noted that the K-means algorithm is sensitive to the selection of initial cluster centers. Several methods have been proposed to select the initial clustering center, notably the improved algorithm K-Means++. K-Means++ Bachem et al. (2016); Arthur & Vassilvitskii (2007) ensures that

the distance between initial centers of mass is maximized, thus reducing the risk of the algorithm converging to a local optimal solution. Lan et al. Lan et al. (2015) proposed initializing cluster centers with density peaks, determining cluster centers based on the local density of data points and their distance to higher density points. Wu et al. Wu et al. (2021) calculated the nearest neighbor density for each point, selected those with the highest density as initial center candidates, and further determined the final initial cluster centers by constructing a minimum spanning tree among these candidates. Additionally, Liao et al. Liao et al. (2024) calculated the decision value for each data point based on the product of the nearest neighbor density peak (NNDP) of data points, and automatically selected the point with the highest decision value as the initial clustering center. Aslam et al. Aslam et al. (2020) evenly divided the data into k partitions by Euclidean distance, using the mean value of each partition as the initial centroid. Laszlo et al. Laszlo & Mukherjee (2006) used a data-based super quadtree and genetic algorithm to select a cluster centroid. Mardi et al. Mardi & Keyvanpour (2021) proposed a genetics-based K-Means (GBKM) algorithm, where the clustering centroid is determined by a genetic algorithm that maximizes the fitness function. Merhad Ay et al. Ay et al. (2023) fixed some clustering centers and then searched for the optimal centers for the remaining clusters.

Although optimizing initial centroids can enhance the stability of K-means, the iterative update process for centroids remains unstable. Consequently, some researchers have proposed variants of the K-means algorithm that avoid direct estimation of the center of mass. Nie et al. Nie et al. (2022) reformulated classical K-means as a trace maximization problem, thus directly assigning each sample to the appropriate cluster without updating the center. Additionally, Pei et al. Pei et al. (2023) introduced k-sum, based on the relationship between spectral clustering and K-means, to avoid estimating the centroid matrix when the number of samples in each cluster is strictly equal. However, these methods often overlook the consistency between data geometries and labels, which can limit their effectiveness on complex datasets.

**Notations**: For clarity and consistency within this document, we introduce the notations used throughout the paper. Scalars are denoted by lowercase letters (e.g., $q$), vectors by bold lowercase letters (e.g., $\mathbf{q}$), and matrices by bold uppercase letters (e.g., $\mathbf{Q}$). The $i$-th row and $j$-th column of matrix $\mathbf{Q}$ are denoted by $\mathbf{q}^i$ and $\mathbf{q}_j$, respectively.

## 3 RETHINKING FOR K-MEANS

The essence of traditional K-means clustering is to find a set of cluster centers such that the sum of the distances between all samples and the cluster centers to which they belong is minimized. Specifically,

$$\min_{\mathbf{m}_j, \mathbf{Y}} \sum_{i,j} y_{ij} \|\mathbf{x}_i - \mathbf{m}_j\|_F^2 \quad \text{s.t.} \quad \mathbf{Y} \in \text{Ind} \tag{1}$$

where $\mathbf{m}_j$ is the $j$-th centroid. The element $y_{ij} = 1$ if sample $\mathbf{x}_i$ belongs to the $j$-th cluster, and $y_{ij} = 0$ otherwise.

Considering the sensitivity of K-means to the selection of initial clustering centers, we introduce Theorem 1. This allows us to reformulate K-means by incorporating the concept of manifold learning, thus better capturing the intrinsic geometry of the data and avoiding the dependence on the centroid matrix.

**Theorem 1** *Let matrices $\boldsymbol{P} = \text{diag}(p_1, \ldots, p_K)$ and $\boldsymbol{H} = \text{diag}(h_1, \ldots, h_N)$, where $p_j = \sum_i y_{ij}$ and $h_i = \sum_j y_{ij}$. Then, we have*

$$\sum_{i,j} y_{ij} \|\boldsymbol{x}_i - \boldsymbol{m}_j\|_F^2 = \sum_{i,l} \|\boldsymbol{x}_i - \boldsymbol{x}_l\|_F^2 s_{il} \tag{2}$$

*where the manifold structure $\boldsymbol{S}$ represents the cluster structure in the data, $\boldsymbol{S} = \boldsymbol{Q}\boldsymbol{Q}^T$, $\boldsymbol{Q} = \boldsymbol{Y}\boldsymbol{P}^{-1/2}$.*

**Proof 1** *Expanding the left side of Eq. (2) yields:*

$$\text{tr} \sum_{i,j} \boldsymbol{x}_i \boldsymbol{x}_i^T y_{ij} - 2\,\text{tr} \sum_{i,j} \boldsymbol{x}_i^T \boldsymbol{m}_j y_{ij} + \text{tr} \sum_{i,j} \boldsymbol{m}_j \boldsymbol{m}_j^T y_{ij} \tag{3}$$

*Taking the partial derivative with respect to $\boldsymbol{m}_j$ and setting it to zero, we find:*

$$\boldsymbol{m}_j = \frac{\sum_i \boldsymbol{x}_i y_{ij}}{p_j} = \boldsymbol{X}\boldsymbol{y}_j p_j^{-1} \tag{4}$$

*Substituting Eq. (4) back into Eq. (3), we see that Eq. (3) simplifies to:*

$$\text{tr} \sum_i \boldsymbol{x}_i \boldsymbol{x}_i^T h_i - \text{tr} \sum_j \boldsymbol{X} \boldsymbol{y}_j p_j^{-1} \boldsymbol{y}_j^T \boldsymbol{X}^T = \text{tr}(\boldsymbol{X}(\boldsymbol{H} - \boldsymbol{Y}\boldsymbol{P}^{-1}\boldsymbol{Y}^T)\boldsymbol{X}^T) \tag{5}$$

*Letting the adjacency matrix* $\boldsymbol{S} = \boldsymbol{Y}\boldsymbol{P}^{-1}\boldsymbol{Y}^T$, *then*

$$\boldsymbol{S}\boldsymbol{1} = \boldsymbol{Y}\boldsymbol{P}^{-1}(\boldsymbol{1}^T\boldsymbol{Y})^T = \boldsymbol{Y}\boldsymbol{1} = \boldsymbol{H}\boldsymbol{1} \tag{6}$$

*Eq. (6) means that* $\boldsymbol{H}$ *is a degree matrix of* $\boldsymbol{S}$*, then Eq. (5) can be written as:*

$$\text{tr}(\boldsymbol{X}(\boldsymbol{H} - \boldsymbol{Y}\boldsymbol{P}^{-1}\boldsymbol{Y}^T)\boldsymbol{X}^T) = \sum_{i,l} \|\boldsymbol{x}_i - \boldsymbol{x}_l\|_F^2 \, s_{il} \tag{7}$$

*Therefore, according to Eq. (3), Eq. (5), and Eq. (7), we can conclude that Eq. (2) holds.* □

In summary, our method constructs the manifold structure $\mathbf{S}$ from label $\mathbf{Y}$, ensuring the consistency of sample labels on the same manifold. At the same time, the estimation of the centroid matrix is avoided. We reinterpret K-means from the perspective of manifold learning to obtain the new form:

$$\min_{\mathbf{Y} \in \text{Ind}} \sum_{i,l} \|\mathbf{x}_i - \mathbf{x}_l\|_F^2 \, s_{il} = \min_{\mathbf{Y} \in \text{Ind}} \sum_{i,l} \|\mathbf{x}_i - \mathbf{x}_l\|_F^2 \left\langle \mathbf{Q}^i, \mathbf{Q}^l \right\rangle = \min_{\mathbf{Y} \in \text{Ind}} \sum_{i,l} d_{il} \left\langle \mathbf{Q}^i, \mathbf{Q}^l \right\rangle$$
$$= \min_{\mathbf{Y} \in \text{Ind}} \text{tr}(\mathbf{Q}^T \mathbf{D} \mathbf{Q}) = \min_{\mathbf{Y} \in \text{Ind}} \text{tr}(\mathbf{Y}^T \mathbf{D} \mathbf{Y} \mathbf{P}^{-1}) \tag{8}$$

where $\mathbf{Y} \in \mathbb{R}^{N \times K}$ denotes the label matrix, and the elements of the distance matrix $\mathbf{D}$ are defined as $d_{il} = \|\mathbf{x}_i - \mathbf{x}_l\|_F^2$, $\mathbf{Q} = \mathbf{Y}\mathbf{P}^{-1/2}$.

## 4 METHODOLOGY

### 4.1 MOTIVATION AND OBJECTIVE

The model (8) is difficult to solve and does not guarantee class equilibrium. Therefore, to optimize the model and ensure the equilibrium of classes after clustering, we introduced Theorem 2 as a solution.

**Theorem 2** *Given* $n_1 + n_2 + \ldots + n_K = N$*, where* $n_j \geq 0$ *represents the number of samples in the* $j$*-th cluster, Eq.(9) reaches its maximum value when* $n_1 = n_2 = \ldots = n_K = \frac{N}{K}$*. In this scenario,* $\boldsymbol{Y}$ *is discrete and exhibits a balanced class distribution.*

$$\max_{\mathbf{Y}} \|\boldsymbol{Y}^T\|_{2,p} \quad s.t. \ \boldsymbol{Y} \geq 0, \boldsymbol{Y}\boldsymbol{1} = \boldsymbol{1} \tag{9}$$

**Proof 2**

$$\left\|\boldsymbol{Y}^T\right\|_{2,p} = \sum_{j=1}^K \|y_j\|_2^p = \sum_{j=1}^K \left(\|y_j\|_2^2\right)^{\frac{p}{2}} = \sum_{j=1}^K a_j^{\frac{p}{2}} \tag{10}$$

*where* $a_j = \|y_j\|_2^2$.

*Let* $\boldsymbol{a} = [a_1, a_2, \ldots, a_K]^T \in \mathbb{R}^{K \times 1}$*,* $\lambda_1 = \lambda_2 = \ldots = \lambda_K = \frac{1}{K}$*.* $f(a_j) = a_j^{\frac{p}{2}}$ *is a convex function with respect to* $a_j$*, then according to Jensen inequality, we have*

$$f\left(\sum_{j=1}^K \lambda_j a_j\right) \geqslant \sum_{j=1}^K \lambda_j f(a_j) = \frac{1}{K} \sum_{j=1}^K f(a_j) = \frac{1}{K}\left\|\boldsymbol{Y}^T\right\|_{2,p} \tag{11}$$

*Equality holds if and only if* $a_1 = a_2 = \ldots = a_K$.

*In order to find the maximum on the right-hand side of the inequality, we can translate to finding the maximum on the left-hand side of the inequality*

$$maxf\left(\sum_{j=1}^K \lambda_j a_j\right) = max(\frac{1}{K}\sum_{j=1}^K a_j)^{\frac{p}{2}} = max(\frac{1}{K}\sum_{j=1}^K \|y_j\|_2^2)^{\frac{p}{2}} = max(\frac{1}{K}\|Y\|_F^2)^{\frac{p}{2}} \tag{12}$$

*We have*

$$\max_{y_{ij} \geq 0, \sum_j y_{ij} = \mathbf{I}} \|\boldsymbol{Y}\|_F^2 = \max_{y_{ij} \geq 0, \sum_j y_{ij} = \mathbf{I}} \sum_{ij} y_{ij}^2 = \max_{y_{ij} \geq 0, \sum_j y_{ij} = \mathbf{I}} \sum_i \sum_j y_{ij}^2 \quad (13)$$

*In Eq.(13), each row of $\boldsymbol{Y}$ is independent, so for each row of $\boldsymbol{Y}$, Eq.(13) becomes*

$$\max_{y_{ij} \geq 0, \sum_j y_{ij} = \mathbf{I}} \sum_{j=1}^{K} y_{ij}^2 \quad (14)$$

*The solution to the maximization problem (14) should be realized when $\mathbf{y}_i$ has only one element equal to 1 and the rest are 0, and the maximum value should be 1. Thus, we can conclude that the problem $(\|Y\|_F^2)^{\frac{p}{2}}$ only reaches a maximum when $\boldsymbol{Y}$ is a discrete matrix.*

*In this case, combined with Eq. (11), we have*

$$f\left(\sum_{j=1}^{K} \frac{1}{K} a_j\right) = f\left(\frac{1}{K} \sum_{j=1}^{K} a_j\right) = f\left(\frac{1}{K} \sum_{j=1}^{K} n_j\right) = f\left(\frac{N}{K}\right) \quad (15)$$

*So we know that when we take the maximum, $a_1 = a_2 = \ldots = a_K = n_1 = n_2 = \ldots = n_K = \frac{N}{K}$.* $\square$

Theorem 2 demonstrates that Eq. (9) can achieve an approximate class equilibrium. Consequently, model (8) is transformed into a continuous model (16) under these constraints.

$$\min_{\mathbf{Y}} tr(\mathbf{Y}^T \mathbf{DY}) - \lambda \|\mathbf{Y}^T\|_{2,p} \quad s.t. \mathbf{Y} \geq 0, \mathbf{Y1} = \mathbf{1} \quad (16)$$

When Eq. (16) achieves the optimal solution, $\mathbf{Y}$ is discrete and each class is balanced.

## 4.2 OPTIMIZATION

The $\ell_{2,p}$-norm, involving the sum of the singular values of a matrix, is generally non-smooth. Therefore, direct optimization of model (16), which incorporates the $\ell_{2,p}$-norm, using gradient descent can be complex and challenging. To simplify the optimization process, we define $f(\mathbf{Y}) = \|\mathbf{Y}\|_{2,p}$ and perform a first-order Taylor expansion at $\mathbf{Y}^{(t)}$ as follows:

$$f(\mathbf{Y}) = f(\mathbf{Y}^{(t)}) + \langle \nabla f(\mathbf{Y}^{(t)}), \mathbf{Y} - \mathbf{Y}^{(t)} \rangle \quad (17)$$

where $\mathbf{Y}^{(t)}$ is the solution at the $t$-th iteration, and $\nabla f(\mathbf{Y}^{(t)})$ is the gradient of $\|\mathbf{Y}\|_{2,p}$.

The derivative of $\|\mathbf{Y}\|_{2,p}$ with respect to $\mathbf{Y}$ is denoted as $\mathbf{H}$, given by:

$$\mathbf{H} = \frac{\partial \|\mathbf{Y}^T\|_{2,p}}{\partial \mathbf{Y}} = p * \mathbf{Y} * diag\left(\frac{1}{\|\mathbf{y}_1\|_2^{2-p}}, \cdots, \frac{1}{\|\mathbf{y}_K\|_2^{2-p}}\right) \quad (18)$$

Ignoring the constant in the Eq.(17), we solve the Eq.(16) iteratively as follows

$$\begin{aligned}
\mathbf{Y}^{(t+1)} &= \underset{\mathbf{Y}}{\operatorname{argmin}} \operatorname{tr}(\mathbf{Y}^T \mathbf{DY}) - \lambda < \nabla f(\mathbf{Y}^{(t)}), \mathbf{Y} > \\
&= \underset{\mathbf{Y}}{\operatorname{argmin}} \operatorname{tr}(\mathbf{Y}^T \mathbf{DY}) - \lambda tr(\mathbf{H}^T \mathbf{Y})
\end{aligned} \quad (19)$$

So we approximate Eq.(16) to Eq.(20), $\mathbf{Y}$ is updated by solving the following problem:

$$\min_{\mathbf{Y1} = \mathbf{1}, \mathbf{Y} \geqslant \mathbf{0}} \operatorname{tr}(\mathbf{Y}^\top \mathbf{DY}) - \lambda \operatorname{tr}(\mathbf{H}^\top \mathbf{Y}) \quad (20)$$

Let $\mathbf{Y} = \begin{bmatrix} \mathbf{y}^i \\ \mathbf{Y}_0 \end{bmatrix}$, $\mathbf{D} = \begin{bmatrix} d_{ii} & \mathbf{d}_{i0}^\top \\ \mathbf{d}_{i0} & \mathbf{D}_0 \end{bmatrix}$, where $\mathbf{Y}_0 \in \mathbb{R}^{(N-1) \times K}$, $\mathbf{d}_{i0} \in \mathbb{R}^{(N-1) \times 1}$, $\mathbf{D}_0 \in \mathbb{R}^{(N-1) \times (N-1)}$. We have:

$$\begin{aligned}
\mathbf{Y}^\top \mathbf{DY} &= \begin{bmatrix} (\mathbf{y}^i)^\top & (\mathbf{Y}_0)^\top \end{bmatrix} \begin{bmatrix} d_{ii} & \mathbf{d}_{i0}^\top \\ \mathbf{d}_{i0} & \mathbf{D}_0 \end{bmatrix} \begin{bmatrix} \mathbf{y}^i \\ \mathbf{Y}_0 \end{bmatrix} \\
&= (\mathbf{y}^i)^\top d_{ii} \mathbf{y}^i + (\mathbf{Y}_0)^\top \mathbf{d}_{i0} \mathbf{y}^i + (\mathbf{y}^i)^\top \mathbf{d}_{i0}^\top \mathbf{Y}_0 + (\mathbf{Y}_0)^\top \mathbf{D}_0 \mathbf{Y}_0
\end{aligned} \quad (21)$$

Let $\mathbf{H} = \begin{bmatrix} \mathbf{h}^i \\ \mathbf{H}_0 \end{bmatrix}$, $\mathbf{H}^\top \mathbf{Y} = \begin{bmatrix} (\mathbf{h}^i)^\top & (\mathbf{H}_0)^\top \end{bmatrix} \begin{bmatrix} \mathbf{y}^i \\ \mathbf{Y}_0 \end{bmatrix} = (\mathbf{h}^i)^\top \mathbf{y}^i + (\mathbf{H}_0)^\top \mathbf{Y}_0$

$$
\begin{aligned}
\mathbf{Y}^\top \mathbf{D} \mathbf{Y} - \lambda \mathbf{H}^\top \mathbf{Y} = {} & (\mathbf{y}^i)^\top d_{ii} \mathbf{y}^i + (\mathbf{Y}_0)^\top \mathbf{d}_{i0} \mathbf{y}^i + (\mathbf{y}^i)^\top \mathbf{d}_{i0}^\top \mathbf{Y}_0 \\
& + (\mathbf{Y}_0)^\top \mathbf{D}_0 \mathbf{Y}_0 - \lambda (\mathbf{h}^i)^\top \mathbf{y}^i - \lambda (\mathbf{H}_0)^\top \mathbf{Y}_0
\end{aligned}
\tag{22}
$$

Then, removing items not related to variable $\mathbf{y}^i$, through the properties of trace operation, we have:

$$
\mathrm{tr}(\mathbf{Y}^\top \mathbf{D} \mathbf{Y} - \lambda \mathbf{H}^\top \mathbf{Y}) = \mathrm{tr}((\mathbf{y}^i)^\top d_{ii} \mathbf{y}^i + 2 \mathbf{y}^i \mathbf{Y}_0^\top \mathbf{d}_{i0} - \lambda \mathbf{y}^i (\mathbf{h}^i)^\top) = \mathbf{y}^i (\mathbf{y}^i)^\top d_{ii} + \mathbf{y}^i \mathbf{g}
\tag{23}
$$

where $\mathbf{g} = 2 \mathbf{Y}_0^\top \mathbf{d}_{i0} - \lambda (\mathbf{h}^i)^\top$.

Thus, the problem of updating the $i$-th row of $\mathbf{Y}$ can be:

$$
\min_{\mathbf{y}^i \mathbf{1} = 1} \mathbf{y}^i (\mathbf{y}^i)^\top d_{ii} + \mathbf{y}^i \mathbf{g}
\tag{24}
$$

As $d_{ii} = 0 (i = 1, 2, \cdots, N)$, (24) can be:

$$
\min_{\mathbf{y}^i} \mathbf{y}^i (2 \mathbf{Y}_0^\top \mathbf{d}_{i0} - \lambda (\mathbf{h}^i)^\top) \Leftrightarrow \min_{\mathbf{y}^i} \mathbf{y}^i (2 \mathbf{Y}^\top \mathbf{d}_i - \lambda (\mathbf{h}^i)^\top)
\tag{25}
$$

$\mathbf{d}_i$ is the $i$-th column of $\mathbf{D}$, $d_{ii} = 0$. $\mathbf{Y}$ denotes the solution before $\mathbf{y}^i$ is updated. Then, the solution of $\mathbf{y}^i$ can be:

$$
y_{ib} = \begin{cases} 1, & b = \arg\min_{j} (2 \mathbf{Y}^\top \mathbf{d}_i - \lambda (\mathbf{h}^i)^\top)_j \\ 0, & \text{otherwise.} \end{cases}
\tag{26}
$$

Algorithm 1 presents the pseudo-code of the optimization procedure.

---

**Algorithm 1:** solve problem (16)

---

1: **Input** distance matrix $\mathbf{D} \in \mathbb{R}^{N \times N}$, cluster number $K$, hyperparameter $\lambda$.
2: **Initialize** label matrix $\mathbf{Y} \in \mathbb{R}^{N \times K}$
3: **repeat**
4:     update matrix $\mathbf{H}$ by Eq. equation 18;
5:     update matrix $\mathbf{Y}$ by Eq. equation 26 row by row;
6: **until** convergence
7: **Output** $\mathbf{Y} \in \mathbb{R}^{N \times K}$

---

### 4.3 COMPUTATIONAL COMPLEXITY ANALYSIS

The time complexity in the optimization method is mainly focused on the solution of $\mathbf{H}$ and $\mathbf{Y}$. Update $\mathbf{H}$ with the equation provided, the computational complexity is $O(NK^2)$. Calculation $\mathbf{Y}^\top \mathbf{D}$ requires multiplying a $K \times N$ matrix with an $N \times N$ matrix, resulting in a time complexity of $O(N^2 K)$. If the outer loop iterates $T$ times, the total complexity of this algorithm becomes $O(T \times (NK^2 + N^2 K))$.

### 4.4 DISTANCE MATRIX

To explore different K-means variants, we can employ various metrics such as the Euclidean distance, KNN distance, or even the Euclidean distance in kernel space. These choices define the distance matrix $\mathbf{D}$. Additionally, we can introduce novel types of distance matrices $\mathbf{D}$ to develop new K-means variants. For instance, using the adjacency matrix $\mathbf{S}$ as described in Lu et al. (2023) offers one such alternative. By applying a suitable transformation function, we can convert the adjacency matrix $\mathbf{S}$ into the distance matrix $\mathbf{D}$ and then construct the anchor graphXia et al. (2023). This conversion incorporates more structural information about the data into the clustering process. The transformation function for our-custom is defined as follows:

$$
d_{ij} = \frac{2}{1 + (2\pi s_{ij})^2}
\tag{27}
$$

The nonlinear transformation specified in Eq.(27) assigns closer distances to similar data points and greater distances to dissimilar ones, enhancing the discriminative power and robustness of the model.

## 5 EXPERIMENTS

We evaluate our proposed model using three toy datasets and nine benchmark datasets. Experiments are conducted on a Windows 10 desktop computer equipped with a 2.40 GHz Intel Xeon Gold 6240R CPU, 64 GB RAM, and MATLAB R2020b (64-bit).

### 5.1 EXPERIMENTS ON ARTIFICIAL DATASETS

In this section, we validate our method for clustering nonlinearly separable data using two synthetic datasets. Specifically, we create a two-moon dataset with 400 samples that form two moon-like shapes on a two-dimensional plane, showcasing nonlinear separability. Additionally, we utilize a three-curve dataset comprising 1,200 samples distributed across three S-shaped curves on a two-dimensional plane, each curve representing a distinct, nonlinearly separable cluster.

The first row of Figure 1 illustrates the clustering effects of the two-moon dataset using various distance measures. Specifically, Figure 1(a) and (b) demonstrate the clustering results based on Euclidean distance, Figure 1(c) depicts the outcome using KNN distance, Figure 1(d) presents the effects using Euclidean distance in kernel space, and Figure 1(e) displays the results achieved by our algorithm using a custom distance measure. The second row in Figure 1 provides a similar comparison for the three-curve dataset. By examining the subfigures in Figure 1, we can clearly see the different clustering effects of various distance measures. Notably, our model effectively utilizes manifold learning techniques combined with a no-center K-means approach to accurately cluster nonlinearly separable data by aligning the data labels with the manifold structure, as demonstrated in Figure 1(e) and (j). This underscores the efficacy of our method in efficiently partitioning nonlinear separable clusters within the input space, thereby enhancing clustering accuracy.

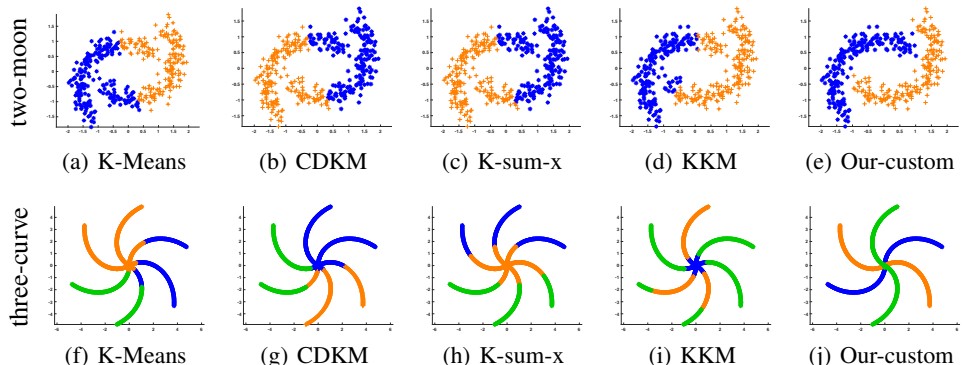

Figure 1: Visualization of artificial datasets. Different colors represent different classes of data.

### 5.2 EXPERIMENTAL ON BENCHMARK DATASETS

#### 5.2.1 DATASETS AND COMPETING ALGORITHMS

We conducted experiments on nine datasets: JAFFE Lyons et al. (1998) consists of 213 expressions from 10 subjects. ORL Samaria & Harter (1994) contains 400 facial images from 40 individuals. UMIST Graham & Allinson (1998) consists of 564 facial images from 20 individuals. Face-V5 [1] consists of 2,500 face images in 500 categories. AR Martinez & Benavente (1998) contains 3,120 face images in 120 classes. isolet[2] contains 7,797 samples of the pronunciation of 26 letters. USPS Hull (1994) consists of 9,298 handwritten digit images. Pendigits [3] is made up of 10,992 handwritten digits. And PEAL Gao et al. (2008) contains 30,863 head and shoulder images of 1,040 people.

---

[1]http://biometrics.idealtest.org/dbDetailForUser.do?id=9

[2]https://archive.ics.uci.edu/dataset/54/isolet

[3]https://odds.cs.stonybrook.edu/pendigits-dataset/

In order to fully evaluate the effectiveness of our proposed method, we selected six clustering algorithms as references for comparative analysis: **K-Means**, **KKM**Tzortzis & Likas (2008), **RKM** Lin et al. (2019), CDKMNie et al. (2022), **K-sum**Pei et al. (2023), **K-sum-x**Pei et al. (2023).

### 5.2.2  RESULTS

**Discussion of the value of p**: To gain a deeper understanding of how different values of parameter $p$ in the $\ell_{2,p}$-norm impact the clustering outcomes, we conducted experiments using our model on the UMIST dataset, as illustrated in Figure 2. We take $p$ between 0.1 and 1, we can find that the overall performance of the model is better when $p = 1$. Therefore, in order to simplify the experiment, we fixed the value as $p = 1$ in this paper.

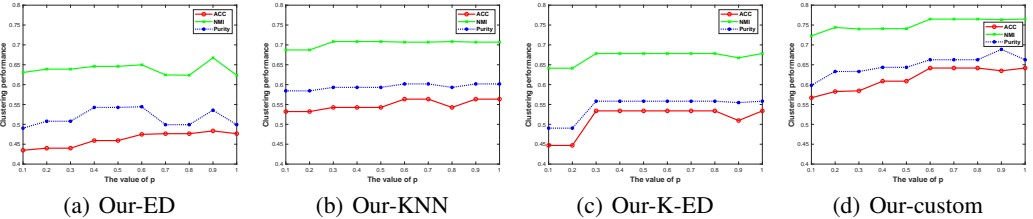

| (a) Our-ED | (b) Our-KNN | (c) Our-K-ED | (d) Our-custom |

Figure 2: Effect of parameter $p$ on UMIST.

After conducting experiments across nine datasets, we obtained the clustering measurement results, as shown in Tables 1 and 2. Based on this analysis, we can draw the following conclusions:

Table 1: The clustering performances on the JAFFE, ORL, UMIST, Face-V5, AR, isolet, USPS, and Pendigits datasets.

| Datasets | JAFFE | | | ORL | | | UMIST | | | Face-V5 | | |
|---|---|---|---|---|---|---|---|---|---|---|---|---|
| Methods | ACC | NMI | Purity | ACC | NMI | Purity | ACC | NMI | Purity | ACC | NMI | Purity |
| K-means | 0.7085 | 0.8010 | 0.7455 | 0.5198 | 0.7234 | 0.5705 | 0.4339 | 0.6410 | 0.5110 | 0.7633 | 0.9369 | 0.8114 |
| KKM | 0.8028 | 0.8246 | 0.8263 | 0.5425 | 0.7440 | 0.5800 | 0.4661 | 0.6682 | 0.5304 | 0.7816 | 0.9380 | 0.8076 |
| RKM | 0.8310 | 0.8159 | 0.8310 | 0.5000 | 0.7143 | 0.5200 | 0.4209 | 0.5963 | 0.4400 | 0.8140 | 0.9473 | 0.8124 |
| CDKM | 0.7451 | 0.8246 | 0.7812 | 0.5507 | 0.7529 | 0.6090 | 0.4210 | 0.6404 | 0.5043 | 0.8506 | 0.9639 | 0.8852 |
| K-sum | 0.8789 | 0.8764 | 0.8789 | 0.6337 | 0.7940 | 0.6562 | 0.4209 | 0.6190 | 0.4553 | 0.9568 | 0.9860 | 0.9656 |
| K-sum-x | 0.8930 | 0.9013 | 0.8977 | 0.5877 | 0.7693 | 0.6060 | 0.4296 | 0.6377 | 0.4715 | 0.9638 | 0.9860 | 0.9662 |
| Our-ED | **0.9671** | 0.9547 | **0.9671** | 0.6575 | 0.8042 | 0.6750 | 0.4765 | 0.6236 | 0.4991 | 0.9684 | 0.9874 | 0.9696 |
| Our-KNN | 0.9484 | 0.9442 | 0.9484 | 0.6575 | 0.8078 | 0.6675 | 0.5635 | 0.7069 | 0.6017 | 0.9724 | 0.9900 | 0.9732 |
| Our-K-ED | **0.9671** | 0.9548 | **0.9671** | 0.6600 | 0.7945 | 0.6700 | 0.5339 | 0.6783 | 0.5583 | **0.9752** | 0.9894 | **0.9760** |
| Our-custom | **0.9671** | 0.9623 | **0.9671** | **0.7050** | **0.8331** | **0.7175** | **0.6348** | **0.7635** | **0.6887** | 0.9724 | **0.9908** | 0.9728 |
| Datasets | AR | | | isolet | | | USPS | | | Pendigits | | |
| Methods | ACC | NMI | Purity | ACC | NMI | Purity | ACC | NMI | Purity | ACC | NMI | Purity |
| K-means | 0.2514 | 0.5574 | 0.2749 | 0.5469 | 0.7154 | 0.5958 | 0.6458 | 0.6026 | 0.7129 | 0.6963 | 0.6705 | 0.7260 |
| KKM | 0.2112 | 0.4786 | 0.2135 | 0.5238 | 0.7029 | 0.5621 | 0.6872 | 0.6437 | 0.7565 | 0.7859 | 0.7139 | 0.7859 |
| RKM | 0.2641 | 0.5752 | 0.3215 | 0.6299 | 0.7346 | 0.6387 | 0.6241 | 0.5748 | 0.7003 | 0.7296 | 0.6639 | 0.7296 |
| CDKM | 0.2653 | 0.5700 | 0.2862 | 0.5328 | 0.7159 | 0.5837 | 0.6526 | 0.6094 | 0.7237 | 0.7027 | 0.6697 | 0.7226 |
| K-sum | 0.2970 | 0.5963 | 0.3686 | 0.6269 | 0.7347 | 0.6402 | 0.6802 | 0.6274 | 0.7486 | 0.7562 | 0.6743 | 0.7562 |
| K-sum-x | 0.2454 | 0.5676 | 0.3236 | 0.6094 | 0.7307 | 0.6254 | 0.6502 | 0.5853 | 0.7150 | 0.7768 | 0.7001 | 0.7768 |
| Our-ED | 0.2612 | 0.5765 | 0.2724 | 0.6538 | 0.7508 | 0.6672 | 0.6539 | 0.5842 | 0.7164 | 0.7816 | 0.7056 | 0.7816 |
| Our-KNN | 0.3359 | 0.6353 | 0.3551 | 0.6504 | 0.7533 | 0.6540 | 0.7545 | 0.6690 | 0.7545 | 0.8406 | 0.7719 | 0.8406 |
| Our-K-ED | 0.2747 | 0.5794 | 0.2865 | 0.6810 | 0.7613 | 0.6930 | 0.7595 | 0.6559 | 0.7595 | **0.8579** | **0.7784** | **0.8579** |
| Our-custom | **0.4333** | **0.7144** | **0.4487** | **0.7454** | **0.8014** | **0.7458** | **0.8450** | **0.7803** | **0.8450** | 0.8322 | 0.7612 | 0.8322 |

**Adaptability to Different Distance Matrices:** Our model can adapt to various types of distance matrices. In the experiments, we compared Our-ED (square Euclidean distance), Our-KNN (KNN distance), Our-K-ED (kernel distance), and Our-custom (custom distance). The experimental results show that the clustering performance of most datasets can be significantly improved by using Our-custom. Specifically, our custom distance employs a method of nonlinear mapping to the adjacency graph, which proves more advantageous than the square Euclidean distance in handling linearly non-separable datasets. The results of kernel distance are comparable to those of KNN distance.

Table 2: The clustering performances on the PEAL dataset.

| Datasets | PEAL | | | | | | | | | |
|---|---|---|---|---|---|---|---|---|---|---|
| Methods | K-means | KKM | RKM | CDKM | K-sum | K-sum-x | Our-ED | Our-KNN | Our-K-ED | Our-custom |
| ACC | 0.7206 | 0.7087 | 0.8072 | 0.7296 | 0.8770 | 0.8491 | 0.8596 | **0.8919** | 0.8602 | 0.8854 |
| NMI | 0.8939 | 0.8624 | 0.9129 | 0.8967 | 0.9424 | 0.9291 | 0.9321 | **0.9417** | 0.9317 | 0.9446 |
| Purity | 0.7539 | 0.7296 | 0.8181 | 0.7617 | 0.8811 | 0.8537 | 0.8640 | **0.8939** | 0.8649 | 0.8889 |

However, the custom distance can more effectively utilize the prior knowledge of the graph and further unearth the intrinsic structural information of the data.

**Evaluation of Clustering Algorithms:** When evaluating the performance of clustering algorithms, we observe that algorithms dependent on the centroid matrix—specifically K-means, KKM, and RKM—perform less effectively on the baseline datasets compared to the K-sum and K-sum-x, which do not require centroid matrix estimation. Specifically, K-sum and K-sum-x combine spectral clustering and K-means in the clustering process, bypassing the need to estimate the centroid matrix. This combination enables them to exhibit higher accuracy when dealing with complex datasets.

**Handling Imbalanced Datasets:** Although K-sum and K-sum-x algorithms initially assume balanced dataset categories, this constraint is removed during the actual solution process. This flexibility may limit their performance in certain scenarios, particularly in datasets with uneven category distribution. In contrast, we reinterpret the K-means algorithm from a manifold learning perspective and subtly incorporate the $\ell_{2,p}$-norm. The integration of the $\ell_{2,p}$-norm not only enhances the model's flexibility but also naturally maintains class equilibrium during the solving process.

### 5.2.3 PARAMETERS SETTING AND ANALYSIS

In order to verify the influence of parameter $\lambda$ on clustering performance, we carry out parametric analysis of the custom distance in our model, as shown in Figure 3. In particular, for the AR dataset, the model with $\lambda = 0.8$ reached the best clustering effect. When facing the USPS dataset, the best clustering performance occurred under $\lambda = 0.6$. For the Pendigits dataset, the performance is optimal when $\lambda = 0.5$. And on the PEAL dataset, the model with $\lambda = 0.9$ showed the best clustering performance. These findings further emphasize the importance of precise adjustment of the $\lambda$ parameters. By fine-tuning these parameters, we can significantly improve the clustering effectiveness of the model, thus obtaining better clustering results on various datasets.

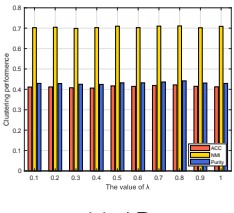 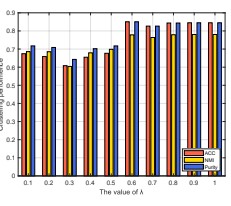 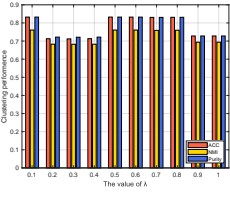 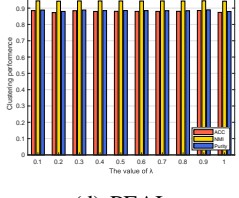

| (a) AR | (b) USPS | (c) Pendigits | (d) PEAL |
|---|---|---|---|

Figure 3: Effect of parameter $\lambda$.

### 5.2.4 TSNE VISUALIZATION

We adopted the TSNE technology to carry out dimensional-reduction processing on several datasets, including JAFFE, USPS, UMIST, and Pendigits. We successfully mapped high-dimensional data to a two-dimensional plane and performed visualization clustering displays, as shown in Figure 4. It can be clearly observed from the figure that the data points are effectively divided into different clusters. The boundaries between the clusters are distinct, and the data points within the clusters are closely adjacent.

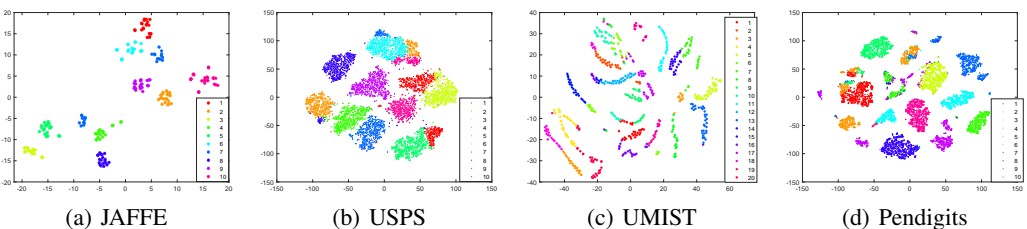

| (a) JAFFE | (b) USPS | (c) UMIST | (d) Pendigits |

Figure 4: TSNE visualization.

### 5.2.5 CONVERGENCE

Our approach is evaluated for convergence and clustering performance on benchmark datasets such as AR, USPS, Pendigits, and PEAL. To quantify the convergence process more accurately, we track the changes in the value of the objective function with the number of iterations. As shown in Figure 5, the value of the objective function tends to converge swiftly. Simultaneously, we use ACC, NMI, and Purity indicators to evaluate the clustering performance of the algorithm. Experimental results show that our method achieves robust clustering performance on these datasets, thus verifying the effectiveness and practicability of the algorithm.

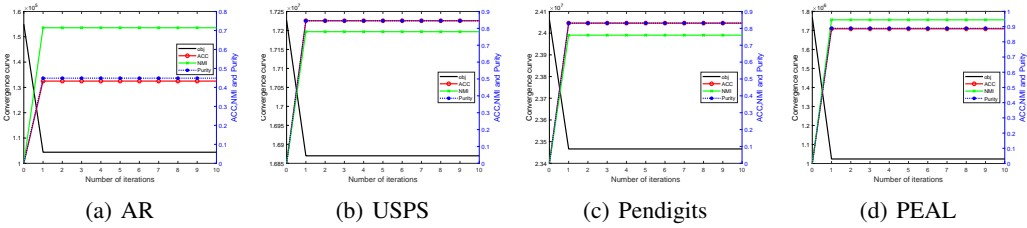

| (a) AR | (b) USPS | (c) Pendigits | (d) PEAL |

Figure 5: Curves of model loss and clustering indexes with number of iterations.

## 6 CONCLUSION

This paper presents a new manifold K-means clustering framework. Different variants of K-means can be obtained by flexibly applying different distance matrices. The framework reconstructs the traditional K-means from the perspective of manifold learning, realizes data clustering without centroid estimation, and ensures the consistency of manifold structure and cluster labels. Additionally, we introduce the maximization of the $\ell_{2,p}$-norm to effectively maintain the class balance in the clustering process. A large number of experimental results fully verify the superiority and effectiveness of this method.

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
