# OpenReview forum: "Manifold K-means with $\ell_{2,p}$-Norm Maximization"
_ICLR.cc/2025/Conference — ICLR 2025 Conference Withdrawn Submission_

### Official Review · Reviewer_zgDc · 2024-10-30

**Soundness:** 2
**Presentation:** 3
**Contribution:** 1
**Rating:** 3
**Confidence:** 5

**Summary:**

The author proposes (1) rewriting $k$-means clustering into a manifold paradigm, (2) utilizing discrete clustering labels  $Y$ , (3) adding $l_{2,p}$-norm constraints to  $Y$  to promote cluster balance, (4) deriving the algorithm and detailing the optimization process, and (5) conducting a series of experiments.

**Strengths:**

1. The article is easy to read.
2. The proofs of two simple theorems are complete and correct.
3. The writing style is generally standard and free of obvious errors.
4. The motivation behind the paper is good, but it falls short of the points mentioned in the contributions by the authors.

**Weaknesses:**

1. Using discrete labels $Y$ is not an innovative work, and many clustering algorithms relax discrete labels to the continuous domain to make the learning process more effective.
2. The author proposes using $l_{2,p}$-norm constraints on the discrete label matrix $Y$ to promote cluster balance. However, (1) applying $l_2$-norm directly could achieve this goal, (2) this is achieved by adding a regularization term to enforce cluster balance, which is not a model-driven effect and can be applied to any clustering algorithm, (3) in Fig.1, only experiments within the range of 0.1 to 1 were performed, without testing values greater than 1.
3. The Pendigits dataset is clearly a dataset with very balanced clusters. When $\lambda$ is high, it should fit this characteristic, yet the clustering performance significantly decreases. Please explain the reason.
4. One of the main contributions claimed by the author is cluster balance; however, no information is provided regarding the characteristics of each cluster in the experimental datasets.
5. From Table 1, it can be seen that the default distance measure used in the comparison algorithms is Euclidean distance without any hyperparameters. However, the proposed algorithm only shows performance improvement on one dataset using Euclidean distance measure when requiring hyperparameter tuning, showing almost no improvement on all other datasets. This suggests that the discretization of $Y$ did not play a substantial role. The improvement in clustering performance mainly stems from changes in the distance measure, which can also be applied to other clustering algorithms.
6. From the loss function and clustering performance convergence plots, it can be observed that convergence is essentially achieved after one iteration. Therefore, in subsequent iterations, does the discrete $Y$ remain unchanged? How can it be proven that the proposed algorithm effectively learns during the optimization process?

**Questions:**

As the weakness.

---

> ### Comment · Reviewer_zgDc · 2024-12-02
> **Thanks for Associate Program Chairs**
>
> Dear Associate Program Chairs,
>
> Thank you for your recognition of our contributions.
>
> Your suggestion is very meaningful, and I will provide more specific recommendations in future reviews.
>
> The authors did not provide any feedback based on my and other reviewers' comments, as well as your suggestions. Therefore, I believe the authors may have abandoned the article.
>
> Best regards.

---

### Official Review · Reviewer_BryM · 2024-10-31

**Soundness:** 3
**Presentation:** 1
**Contribution:** 3
**Rating:** 5
**Confidence:** 5

**Summary:**

In literature, k-means is limited to the sensitivity to the selection of the initial cluster center and nonlinear separable data. To overcome the two issues, the authors draw inspiration from manifold learning and propose a manifold K-means clustering framework.

**Strengths:**

1. The proposed method seems effective according to experiment results of Table 1.

**Weaknesses:**

1. In line 51-52, the authors point that some methods overlook the alignment of data geometry with labels. Could the authors explain this more clearly?

2. The proposed objective is Eq. (16) which is similar to the kernel k-means objective (can find in Section 2.1 of [1]). Could the authors discuss their similarities and differences?

3. The first two contributions in Introduction are duplicated, so do the last two.

4. In table 2, KNN is used to construct the distance matrix. Since KNN is a supervised method and this paper focus on clustering, the proposed 'our-KNN' method should be further clarified.

5. Currently, the words in figures are too small and should be enlarged.

6. The expression should be polished. For example, "equation 18" in line 299 should be "Equation (18)" or "Eq. (18)", etc.

7. Convergence analysis should be provided.

[1] Liu J, et al. Optimal Neighborhood Multiple Kernel Clustering with Adaptive Local Kernels, TKDE.

**Questions:**

Please see Weakness

---

### Official Review · Reviewer_LkpW · 2024-11-01

**Soundness:** 2
**Presentation:** 3
**Contribution:** 2
**Rating:** 5
**Confidence:** 4

**Summary:**

This paper proposes a manifold k-means method, which reformulates the k-means to the manifold learning, and then plugged a balanced regularized term into it. The experimental results show its effectiveness.

**Strengths:**

1. The experimental results are good.
2. The presentation is good and easy to follow.

**Weaknesses:**

1. My major concern is about its novelty. The relationship between k-means and manifold learning or spectral clustering has been widely studied in previous works, e.g. [1] and [2]. Section 3 (i.e., Rethinking for K-means) of this paper seems not to provide any new insight about the k-means. In addition, the balance regularized term seems a simple extension of previous works. Previous works, e.g.[3], show that when $p=1$, it can lead to a balanced result. This paper seems only an extension of $p$, which is not significant enough for publication in ICLR.

2. Since the paper proposes a balance regularized term, in the experiments, they should also compare with some state-of=the-art balanced clustering methods to show the effectiveness.

3. I'm also interested in the case that the ground truth is imbalanced. If in this case, how does the proposed method perform? It would be better to conduct the experiments to discuss this case.

[1] Centerless Clustering, in IEEE TPAMI 2023.
[2] Efficient Clustering Based On A Unified View Of K-means And Ratio-cut, in NeurIPS 2020.
[3] Balanced k-Means and Min-Cut Clustering, in Arxiv 2014.

**Questions:**

Please see Weaknesses.

---

### Official Review · Reviewer_9cF4 · 2024-11-02

**Soundness:** 2
**Presentation:** 2
**Contribution:** 2
**Rating:** 5
**Confidence:** 4

**Summary:**

This paper introduces a new manifold-based K-means clustering framework that addresses limitations in standard K-means, particularly sensitivity to initial cluster centers and difficulty with nonlinearly separable data. By leveraging manifold learning, the authors redefine K-means to process complex geometric structures directly, without the need to compute a centroid matrix. The paper also introduces the ℓ2,p-norm, which is maximized to balance class distributions within the clustering process. Extensive experiments and theoretical analyses demonstrate that the proposed manifold K-means method outperforms traditional and kernel-based K-means on various datasets.

**Strengths:**

1.The paper presents an approach by redefining K-means in a manifold framework, leveraging the ℓ2,p-norm to balance class distributions.
2.Methodology and theoretical basis are well-structured, with explanations supporting the novel aspects of the method.
3.The approach has potential in clustering applications that require handling complex, nonlinear data.

**Weaknesses:**

1.The paper does not fully clarify how the proposed method theoretically differs or improves upon existing manifold-based clustering methods.
2.While the method is promising, the lack of real-world applications or case studies makes it difficult to gauge its practical impact.
3.Experiments are limited to controlled datasets, leaving questions about performance and scalability in high-dimensional, real-world scenarios.
4.Some terminology and symbols lack clarity, especially in the theoretical sections, which may hinder readability.

**Questions:**

1.How does the computational complexity of this method compare to traditional and kernel-based K-means?
2.Has a sensitivity analysis been performed on the ℓ2,p-norm parameter? Does the model perform consistently across parameter variations?
3.How does the manifold structure improve clustering accuracy or interpretability, compared to other manifold clustering methods?
4.Could the authors provide evidence supporting the claim that the method maintains consistency between manifold structure and clustering labels?
5.What are the specific advantages of this approach over K-means++ or other initialization-robust methods in complex clustering tasks?
6.Have the authors validated the scalability of this approach on higher-dimensional, real-world datasets, where nonlinear structures may be more intricate?

---

### Note · Authors · 2025-01-23

I have read and agree with the venue's withdrawal policy on behalf of myself and my co-authors.